# Water Information Extraction Based on Multi-Model RF Algorithm and Sentinel-2 Image Data

**Zhiqi Jiang [1]**, **Yijun Wen [2,\*]**, **Gui Zhang [3]** and **Xin Wu [4]**

1   School of Forestry, Central South University of Forestry and Technology, Changsha 410004, China; jiangzhiqi2022@163.com
2   Faculty of Science, Central South University of Forestry and Technology, Changsha 410004, China
3   National Forest Fire Prevention Virtual Simulation Laboratory Teaching Center, Changsha 410004, China; csfu3s@163.com
4   Key Laboratory of Digital Dongting Lake of Hunan Province, Changsha 410004, China; csuftfire@csuft.edu.cn
\*   Correspondence: wenyishalong@163.com; Tel.: +86-18207370571

**Abstract:** For the Sentinel-2 multispectral satellite image remote sensing data, due to the rich spatial information, the traditional water body extraction methods cannot meet the needs of practical applications. In this study, a random forest-based RF_16 optimal combination model algorithm is proposed to extract water bodies. The research process uses Sentinel-2 multispectral satellite images and DEM data as the basic data, collected 24 characteristic variable indicators (B2, B3, B4, B8, B11, B12, NDVI, MSAVI, B5, B6, B7, B8A, NDI45, MCARI, REIP, S2REP, IRECI, PSSRa, NDWI, MNDWI, LSWI, DEM, SLOPE, SLOPE ASPECT), and constructed four combined models with different input variables. After analysis, it was determined that RF_16 was the optimal combination for extracting water body information in the study area. Model. The results show that: (1) The characteristic variables that have an important impact on the accuracy of the model are the improved normalized difference water index (MNDWI), band B2 (Blue), normalized water index (NDWI), B4 (Red), B3 (Green), and band B5 (Vegetation Red-Edge 1); (2) The water extraction accuracy of the optimal combined model RF_16 can reach 93.16%, and the Kappa coefficient is 0.8214. The overall accuracy is 0.12% better than the traditional Relief F algorithm. The RF_16 method based on the optimal combination model of random forest is an effective means to obtain high-precision water body information in the study area. It can effectively reduce the "salt and pepper effect" and the influence of mixed pixels such as water and shadows on the water extraction accuracy.

**Keywords:** random forest; Sentinel-2; red-edge remote sensing data; water extraction

## 1. Introduction

As an important part of the earth's water cycle, land surface water, such as rivers, lakes, and reservoirs, are irreplaceable for the global ecosystem and climate system [1]. Surveying land surface water bodies and delineating their spatial distribution are meaningful to the understanding of hydrology processes and the management of water resources [2]. Remote sensing monitoring has become an important method for monitoring surface water resources in recent years. Compared with traditional field survey methods, it has great advantages of macroscopicity, instantaneity, dynamism, and cost effectiveness.

With the continuous development of research technology, the application of machine learning to various disciplines is becoming more and more extensive, such as support vector machine (SVM) [3,4], random forest (RF) [5–7], artificial neural networks (ANNs) [8,9], classification and regression tree (CART) [10,11], and so on. There are also many machine learning algorithms based on different applications, such as that used by Shaban and other researchers based on multi-objective optimization algorithms to predict the compressive strength of pozzolanic materials RAC [12]. Among them, the random forest classification algorithm demonstrates its outstanding ability in dealing with large data sets, and it also

shares strong anti-noise and anti-overfitting abilities and shows sound effect in classification [13]. Therefore, more and more scholars apply the random forest machine learning method to remote sensing image classification and terrain characteristic extraction research, for example, Zheng Xiaorou et al. used the random forest algorithm, CART decision tree algorithm, and support vector machine to classify land use. Comparing the classification results of several algorithms, it was found that the classification results of the random forest algorithm were more accurate [14]; using Sentinel-1 and Sentinel-2 data, Tufail Rahat and others used a machine learning random forest classification algorithm to classify land use categories, and the results showed an overall accuracy of 97% and Kappa coefficient of 0.97 [15]; VF Rodriguez-Galiano et al. applied the random forest algorithm to land cover classification in a complex area to classify 14 different land classes in southern Spain. The results show that the algorithm obtains accurate land cover classification results with an overall accuracy of 92% and a Kappa coefficient of 0.92. RF is robust to noise reduction and noise in training data [16]. Raziye Hale TOPALOĞLU and other researchers compared and analyzed classification accuracies of land cover/use maps created from Sentinel-2 and Landsat-8 data. According to overall classification accuracies, it can be seen that, for both MLC and SVM classification methods, the image classification accuracies of the Sentinel-2 dataset is better than the Landsat-8 dataset. Although the Sentinel-2 dataset resampled to 30 m using Landsat-8 as a base, the overall classification accuracies of Sentinel-2 are much higher than Landsat-8 data [17]; in addition, a large number of studies have shown that the introduction of terrain factors can significantly improve the classification accuracy of land use or water body extraction [18,19].

Forkuor and other researchers compared and explored the synergistic use of Landsat-8 and Sentinel-2 data in mapping land use and land cover (LULC) in rural Burkina Faso. Specifically, the contribution of the red-edge bands of Sentinel-2 in improving LULC mapping was examined. It was found that the classification of the Sentinel-2 red-edge bands alone produced better and comparable results to Landsat-8 and the other Sentinel-2 bands, respectively. The results of this study demonstrate the added value of the Sentinel-2 red-edge bands and encourage multi-sensorial approaches to LULC mapping in West Africa [20]. Elhadi Adam and Hristos Tyralis used the RF algorithm to study the land use classification of cultivation areas and mountain areas [21,22], and the accuracy of the results exceeded 80%. Swapan Talukdar and Pankaj Singha examined six machine learning algorithms, namely random forest (RF), support vector machine (SVM), artificial neural network (ANN), fuzzy adaptive resonance theory-supervised predictive mapping (Fuzzy ARTMAP), spectral angle mapper (SAM), and Mahalanobis distance (MD). Accuracy assessment was performed using Kappa coefficient, receiver operational curve (RoC), index-based validation, and root mean square error (RMSE). Results of the Kappa coefficient show that all the classifiers have a similar accuracy level with minor variation, but the RF algorithm has the highest accuracy at 0.89 and the MD algorithm (parametric classifier) has the least accuracy at 0.82. Finally, this review concludes that the RF algorithm is the best machine learning LULC classifier among the six examined algorithms [23]; in addition, a large number of studies have shown that the introduction of terrain factors can significantly improve the efficiency of land use and the water extraction classification accuracy [24,25].

In the study of water information extraction, compared with other remote sensing data, Sentinel-2A image has a higher extraction accuracy. For example, the three multispectral bands of the Spot5 satellite have a spatial resolution of 10 m and a temporal resolution of 26 days. Despite having high spatial resolution, the satellite has a small number of bands and a limited application field. Although it can meet the needs of water extraction, the data acquisition of it is paid with a high price of image, which cannot be widely used by most researchers. Landsat-8 has a re-entry period of 16 days and a spatial resolution of 30 m in the visible band. Although it has seven spectral bands, a wide range of applications, and free access to a large amount of data, LandSat8′s spatial resolution has some limitations [26]. The temporal resolution of the Gaofen-1 satellite reaches 4 days, and its spatial resolution of free data is 16 m. Sentinel-2A has a single-star re-entry period of 10 days, and 5 days

for double-star; its spatial resolution of multispectral four bands is 10 m. In contrast, Sentinel-2A has considerable spatial and temporal resolution with free data acquisition. It has great application prospects in water extraction. At the same time, compared with traditional remote sensing data, Sentinel-2A remote sensing data adds four red-edge bands (B5, B6, B7, B8A) that are closely related to chlorophyll content. These bands are used in land use cover classification, wetland extraction, and crops. The research on identification has achieved good results.

Although the Sentinel-2A image has its own data advantages, due to the rich spatial information of the Sentinel-2 multispectral satellite image remote sensing data, the traditional water extraction methods cannot meet the needs of practical applications. In this paper, a random forest-based RF_16 optimal combination model algorithm is proposed to extract water bodies. In the research process, Sentinel-2 multispectral satellite images and DEM data were used as the basic data, and four kinds of variables were collected, including traditional remote sensing data, red-edge remote sensing data, water body index factor, and terrain factor; a total of 24 characteristic variable indicators (B2, B3, B4, B8, B11, B12, NDVI, MSAVI, B5, B6, B7, B8A, NDI45, MCARI, REIP, S2REP, IRECI, PSSRa, NDWI, MNDWI, LSWI, DEM, SLOPE, SLOPE ASPECT) using these four combined models with different input variables were constructed using characteristic indicators, and the influence and contribution of each variable to the extracted water body were analyzed at the same time. After analysis, it was determined that RF_16 is the optimal combination model for extracting water body information in the study area.

## 2. Materials and Methods

### 2.1. Study Area

Xiangtan City is located in the central eastern part of Hunan Province, with a total area of 5015 km$^2$, spanning 111°58′–113°05′ E, 27°20′55″–28°05′40″ N [27], with a subtropical warm and humid climate, and vegetation dominated by plantation forests. The city's main roads are in the shape of a ring, the outer ring mainly consisting of Xianglian Avenue, Furong Avenue, East Erhuan Road, East Jiuzhao Road, West Jiuzhao Road, East Jinpeng Road, Baoshui Road, Shimatou Road, 107 Provincial Road, Shanghai-Kunming Expressway, Xuchang-Guangzhou Expressway, West Dapeng Road, and Xiangzhang Road; the inner ring is composed of Hutan Road, North Jianshe Road, South Shuangyong Road, Middle Shuangyong Road, and Qingyuan Road. According to the administrative boundaries, the area can be further divided into Yuhu District, Xiangtan County, and the rural-urban fringe in Yuetang District (see Figure 1). The study area belongs to a typical subtropical warm and humid climate zone, with distinct seasons and rich rainfall. The winter here is cold, and summer is hot.

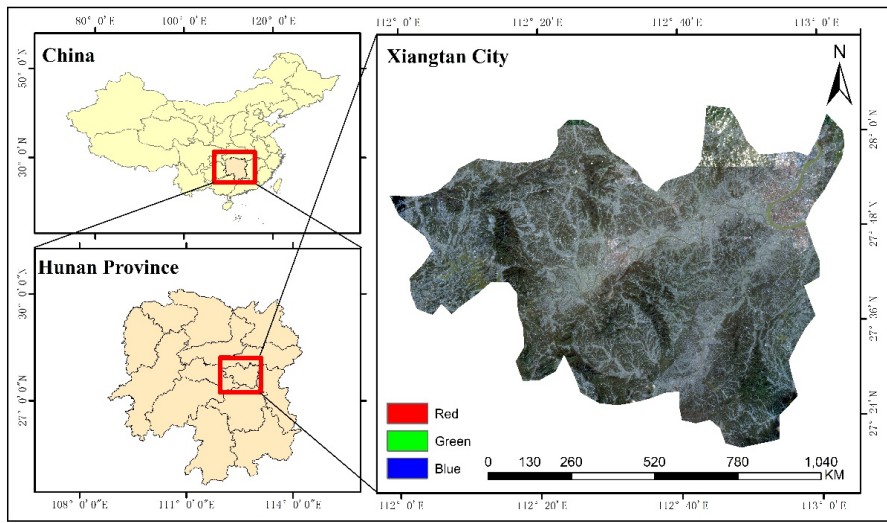

**Figure 1.** Locations of the study areas.

### 2.2. Data Sources and Pre-Processing

Xiangtan City has distinct seasons, and the higher quality images are mainly concentrated in summer and autumn from July to August. Therefore, the study selects nine scenes of Sentinel-2 image data on 3 July 2020, 10 July 2020, 17 July 2020, 24 July 2020, 29 July 2020, 5 August 2020, 12 August 2020, and 19 August 2020. The Sentinel-2 data can be downloaded for free from the ESA website (https://scihub.copernicus.eu/dhus/#/home). Other supporting data in the study include: 30 m resolution DEM products, DEM data downloaded from Geospatial Data Cloud (https://scihub.copernicus.eu/dhus/#/home), and 2018 Xiangtan city land use status vector data (http://www.globallandcover.com/) terrain factors (including elevation, slope, and slope aspect). Through DEM extraction, the current land use status map and Google Earth data were used for the collection and accuracy evaluation of the land class sample points.

Data Pre-Processing

Sentinel-2 data pre-processing mainly includes atmospheric correction, image registration, and image fusion. Since the downloaded Sentinel-2 L1C image is an atmospheric apparent reflectance product that has been geometrically corrected, the SNAP software plug-in Sen2Cor-2.4.0 is used to perform atmospheric correction to obtain the real reflectance data of the ground objects. SNAP software is available for free download through ESA's (European Space Agency) data distribution website (https://scihub.copernicus.eu/). Then, the bilinear resampling was performed in SNAP, and the original 60 m resolution band data was removed. The Sentinel-2-related band parameters used are shown in Table 1. The data come from the following website: https://eros.usgs.gov/sentinel-2. Finally, the author uses ENVI's "Seamless Mosaic" tool for stitching and ENVI's "Subset Data from ROIs" tool for cropping to obtain the image data of Xiangtan City.

**Table 1.** Sentinel-2 Data Band Parameters.

| Sentinel-2 Bands | Center Wavelength (nm) | Spectral Width (nm) | Spatial Resolution (m) |
|---|---|---|---|
| Band 1-Coastal aerosol | 443 | 20 | 60 |
| Band 2-Blue | 490 | 65 | 10 |
| Band 3-Green | 560 | 35 | 10 |
| Band 4-Red | 665 | 30 | 10 |
| Band 5-Vegetation Red-Edge 1 | 705 | 15 | 20 |
| Band 6-Vegetation Red-Edge 2 | 740 | 15 | 20 |
| Band 7-Vegetation Red-Edge 3 | 783 | 20 | 20 |
| Band 8-NIR | 842 | 115 | 10 |
| Band 8a-Vegetation Red-Edge 4 | 865 | 20 | 20 |
| Band 11-SWIR1 | 1610 | 90 | 20 |
| Band 12-SWIR2 | 2190 | 180 | 20 |

## 3. Study Methods and Steps

### 3.1. Study Methods

3.1.1. Random Forest Algorithm

The random forest algorithm (RF algorithm) is a machine learning algorithm based on the decision tree combination composition proposed by Breiman. It is suitable for processing high-dimensional data and is not easy to produce overfitting. The basic principle of the algorithm is as follows:

(1) Through the Bootstrap Method with resampling, N groups of Bagging are randomly selected from the original data set with replacement. The size of each Bagging is about 2/3

of the original data; and the size of the test data set is about 1/3 of the original data, which is called out of bag (OOB) data.

(2) Then, according to the minimum principle of the Gini Coefficient, N groups of Bagging are randomly selected to form N decision trees, and the subset of each node variable after internal splitting is used to construct multiple CART decision trees and form a random forest. The definition formula of the Gini Coefficient is shown in Equation (1), where $T$ is a given data set, $C_i$ is a sample randomly selected and identified as a certain category, and $f(C_i, T)/|T|$ is the probability that the selected sample is $C_i$.

$$Gini = \sum\sum_{i \neq j}(f(C_i, T)/|T|)(f(C_j, T)/|T|) \tag{1}$$

In the equation, $T$ is a given data set, $C_i$ is a sample randomly selected and identified as a certain category, and $f(C_i, T)/|T|$ represents the probability that the selected sample is $C_i$.

(3) The generated random forest classifier classifies the data. For the accuracy estimation, when each sample belongs to the OBB sample, its votes are counted each time, and the votes of the majority will decide the classification category. Since OBB samples are not involved in the establishment of decision trees, they can be used to estimate prediction errors, so as to use OBB errors to evaluate model performance and quantify the importance of each characteristic variable [28]. The importance discrimination of characteristic variables can improve the accuracy of water extraction. The importance of variables is defined in Equation (2):

$$V\left(k^j\right) = \frac{1}{N}\sum_{i=1}^{t}\left(e_i^j - e_i\right) \tag{2}$$

In the Equation (2), $V\left(k^j\right)$ is the importance of j characteristic variables, $N$ is the generated decision tree, $e_i$ is the OOB error of the i-th decision tree, and $e_j^i$ is the new OOB error calculated after randomly changing the j-th characteristic variable.

### 3.1.2. Accuracy Verification Algorithm

Accuracy evaluation is an effective means of judging the quality of water extraction methods, so as to determine the water extraction methods suitable for the study area. The accuracy evaluation method usually adopts the confusion matrix. The mathematical expression equation of confusion matrix is shown in (3):

$$C_M = \begin{bmatrix} x_{11} & x_{12} & \dots & x_{1n} \\ x_{21} & x_{22} & \dots & x_{23} \\ \vdots & \vdots & \ddots & \vdots \\ x_{n1} & x_{n2} & \dots & x_{nn} \end{bmatrix} \tag{3}$$

where the $x_{ij}(i, j = 1, 2, \dots n)$ element represents the probability that the training sample is divided into class J, the i-th row of the matrix represents the probability that the i-th type of training sample is identified as other types, obviously $\sum_{j=1}^{n} x_{ij} = 1$, while the j-th column represents the probability that each type of training sample is identified as class j by the sensor, and the diagonal element represents the probability that the class $i$ sample in the training sample is accurately identified.

The Kappa coefficient is a statistical method to determine the classification accuracy, and it is a commonly used parameter in the consistency test. The closer the Kappa value is to 1, the better the representation, and vice versa, the worse the representation.

$$\text{Kappa} = \frac{N.\sum_i^r x_{ij} - \sum(x_{i+} \cdot x_{+i})}{N^2 - \sum(x_{i+} \cdot x_{+i})} \tag{4}$$

where $r$ is the number of rows of the error matrix; $x_{ij}$ is the value on the main diagonal; $x_{i+}$ and $x_{+i}$ is row $i$ and column $j$, respectively, and $N$ is the total number of samples.

$$Q = \frac{C}{A} \tag{5}$$

where $Q$ is the overall classification accuracy, $C$ is the exact number of pixels, and $A$ is the total.

### 3.2. The Extraction of Characteristic Variables

Based on Sentinel-2 image data and terrain data, this study extracted 24 characteristic indicators. The spectral band data included B2-B8A, B11, and B12 bands, and the role of the red-edge band in land use classification was taken into consideration [29]. The authors divide it into 4 types: (1) traditional remote sensing data: B2, B3, B4, B8, B11, B12, NDVI, and MSAVI; (2) red-edge remote sensing data: B5, B6, B7, B8A, NDI45, MCARI, REIP, S2REP, IRECI, and PSSRa; (3) water indices factors: NDWI, MNDWI, and LSWI; and (4) terrain factors: elevation, slope, and slope aspect. The relevant calculation equation and indices descriptions are shown in Table 2. The spectral reflectance curve of ground objects corresponding to each characteristic variable is shown in Figure 2. Considering the important role of the red-edge band in land use classification, these spectral characteristic indices are divided into four categories. At the same time, in order to test the classification effect of various variables, four models are constructed in the study: (1) Model A: traditional remote sensing data; (2) Model B: traditional remote sensing data + red-edge remote sensing data; (3) Model C: traditional remote sensing data + red-edge remote sensing data + water indices factor; and (4) Model D: traditional remote sensing data + red-edge remote sensing data + water indices factor + terrain factor. By comparing and analyzing the importance ranking of characteristic variables, the author selects the best classification scheme.

**Table 2.** Characterization of spectral indices.

| Abbreviations | Full Names | Calculation Formulas | Types | Citations |
|---|---|---|---|---|
| NDVI | Normalized Difference Vegetation Indices | (B8 − B4)/(B8 + B4) | Traditional Spectral Indices Characteristics | [30] |
| LSWI | Land Surface Water Indices | (B8 − B11)/(B8 + B11) | Traditional Spectral Indices Characteristics | [31] |
| NDWI | Normalized Difference Water Indices | (B8 − B4)/(B8 + B4) | Traditional Spectral Indices Characteristics | [32] |
| MSAVI | Modified Soil Adjusted Vegetation Indices | 0.5 × (2 × (B8+1) − sqrt((2 × B8 + 1)2 − 8 × (B8 − B4))) | Traditional Spectral Indices Characteristics | [33] |
| MNDWI | Modified Normalized Difference Water Indices | (B3 − B11)/(B3 + B11) | Traditional Spectral Indices Characteristics | [34] |
| NDI45 | Normalized Difference Indices | (B5 − B4)/(B5 + B4) | Red-Edge Spectral Indices Characteristics | [35] |
| MCARI | Modified Chlorophyll Absorption Ratio Indices | [(B5 − B4) − 0.2 × (B5 − B3)] × (B5 − B4) | Red-Edge Spectral Indices Characteristics | [36] |
| REIP | Red-Edge Inflection Point Indices | 700 + 40 × ((B4 + B7)/2 − B5)/(B6 − B5) | Red-Edge Spectral Indices Characteristics | [37] |
| S2REP | The Sentinel-2 Red-Edge Position Indices | 705 + 35 × ((B4 + B7)/2 − B5)/(B6 − B5) | Red-Edge Spectral Indices Characteristics | [38] |
| IRECI | Inverted Red-Edge Chlorophyll Indices | (B7 − B4)/(B5/B6) | Red-Edge Spectral Indices Characteristics | [39] |
| PSSRa | Pigment Specific Simple Ratio(chlorophyll) Indices. | B7/B4 | Red-Edge Spectral Indices Characteristics | [40] |

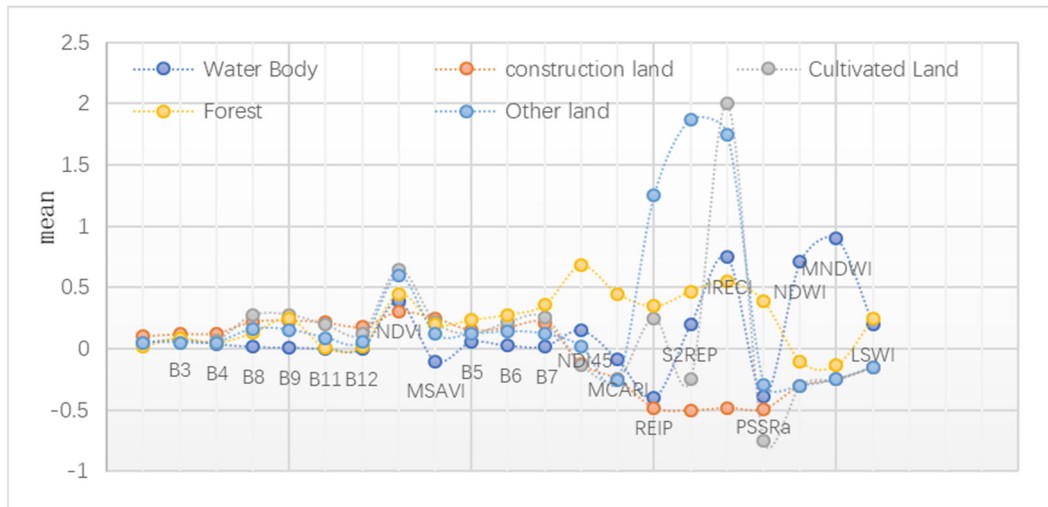

**Figure 2.** Spectral curve of surface features.

*3.3. Sample Selection*

The study combines the vector data and Sentinel-2 data of Xiangtan City's land use status in 2018 to collect land samples within its boundaries. To ensure the scientificity and accuracy of the study and to make the sample fully satisfy the characteristics of random distribution, the authors used the following approach for sample selection: (1) since the Sentinel-2 data has been resampled to a spatial resolution of 10 m, ArcGIS 10.4 software is used to generate a grid of 30 m × 30 m; (2) perform spatial overlay analysis on grid and current land use vector data, and filter attribute data with an area equal to 100 m$^2$; (3) vector data conversion was performed to convert the surface data into point data, and a total of about 300,000 points were extracted. On this basis, in order to improve the efficiency of random forest, the authors selected 10,000 sample points by ArcGIS random sampling tool. The selected sample points adequately cover the study area and meet the requirement of uniform distribution. In order to further ensure the accuracy of the data, the author imported the samples into Google Earth and eliminated outliers. There are 7226 samples finally retained, including 1390 water areas, 1849 cultivated land areas, 2987 forest land areas, 219 grass land areas, 156 wetlands, 249 other land areas, and 376 construction land areas.

The samples are divided into two parts, 70% of them are used as the training sample set for modeling, and the remaining 30% are used as the validation set and do not participate in the model construction. Additionally, they are used to evaluate the classification effect of each model. The number of sample points for each object classification is shown in Table 3. The classification accuracy is evaluated by the overall accuracy (OA), Kappa coefficient, producer's accuracy (PA), and user's accuracy (UA). Among them, Kappa coefficient is an index to measure the classification accuracy, which can measure the consistency between the classification results of the model and the true results at the same time. Its value ranges from −1 to 1, and the closer it is to 1, the more consistent the two are; when it is close to 0, it indicates that the two are consistent with each other as expected by chance, which means the consistency is poor; when it is close to −1, it indicates that the consistency between the two is very low, which means the consistency is extremely poor.

**Table 3.** Number of samples for each class.

| Characteristic Class | Training Samples | Validation Samples | Total Number of Samples |
|---|---|---|---|
| Water | 927 | 463 | 1390 |
| Cultivated land | 1233 | 616 | 1849 |
| Forest land | 1996 | 991 | 2987 |
| Grass land | 146 | 73 | 219 |
| Wetland | 104 | 52 | 156 |
| Construction land | 251 | 125 | 376 |
| Other land | 166 | 83 | 249 |

## 4. Results

### 4.1. Results of Classification and Accuracy Evaluation

In order to improve the accuracy of classification, this study adopts a grid-search based on OOB error value. This method employs ntree and mtry to search for parameter optimization. By employing ntree and mtry, grid-search uses M and N values. By multiplying M and N, it disciplines different RF classifiers and estimates its learning accuracy according to OOB error value. In this way, the exact combination with the highest learning accuracy is regarded as the optimal parameter, whose advantage is to assure the optimal solution in the delineated grid and avoid significant errors. The RF classification algorithm and grid-search algorithm are both implemented in Matlab 2012B language platform. The grid-search allows us to search for the optimal parameter according to the RF algorithm for the dour models. The scope of mtry parameter optimization for model A is (2, 3, 4, 5); for model B it is (3, 5, 7, 9); for model C it is (5, 10, 15); and for model D it is (6, 12, 18). The scope of ntree parameter optimization for all models is unified as (25, 50, 75, 100). The mtry and ntree optimal parameters for the four models are, respectively, (4, 100), (7, 100), (10, 100), and (12, 100). The above parameters are used to execute RF classification algorithm, and the results are shown in Figure 3. Figure 4 is a schematic diagram of water areas superimposed by a false color image. The overall accuracy of the four models are, respectively, 81.24%, 86.39%, 89.71%, and 92.37%. With the increasing of parameters, the overall accuracy and Kappa coefficient increase accordingly. The results show that the addition of terrain factors and red-edge remote sensing data can improve the accuracy of classification recognition. The conclusions are in line with those of Forkuor [29] and A. Htitiou [40]. Figure 3 is accuracy comparison of four models (model A, model B, model C, and model D).

From Table 4, we can see that as the parameter increases, the classification accuracy varies for different land. Therefore, the figure of local area classification results is selected. Three regions of the figure will be selected randomly so as to make a specific contrast analysis. Figure 4 is the standard false color image of Sentinel-2 data in Xiangtan City. Figure 4a–c are images of three local areas in Xiangtan City. Figures 5–7 are classification images corresponding with three local areas. According to the contrast of the three figures, we can see that the classification accuracy of model A is lower than that of model B. The minor error is mainly due to the wrong classification of wetland and cultivation land. Based on the traditional remote sensing image data, model B employs the red-edge remote sensing data B5, B6, B7, B8A, NDI45, MCARI, REIP, S2REP, IRECI, and PSSRa. Therefore, its producer's accuracy has increased 5.47%, and the user's accuracy increases 6.89%. Meanwhile, red-edge remote sensing data can effectively decrease the influences of soil background noise, making the water extraction accuracy increase from 80.63% to 82.16%. Despite this, the accuracy enhancement is not noticeable. The reason is that the red-edge band can more accurately identify different kinds of vegetation. According to the contrast between A1 and D1 of Figure 5, A2 and D2 of Figure 6, and A3 and D3 of Figure 7, we can see clearly that the black and white dots in model A are fewer than those of model D. The whole figure is clearer and prettier. However, the producer's accuracy and user's accuracy of building land have decreased 1.62% and 2.2%, respectively. It might be the image by red-edge band.

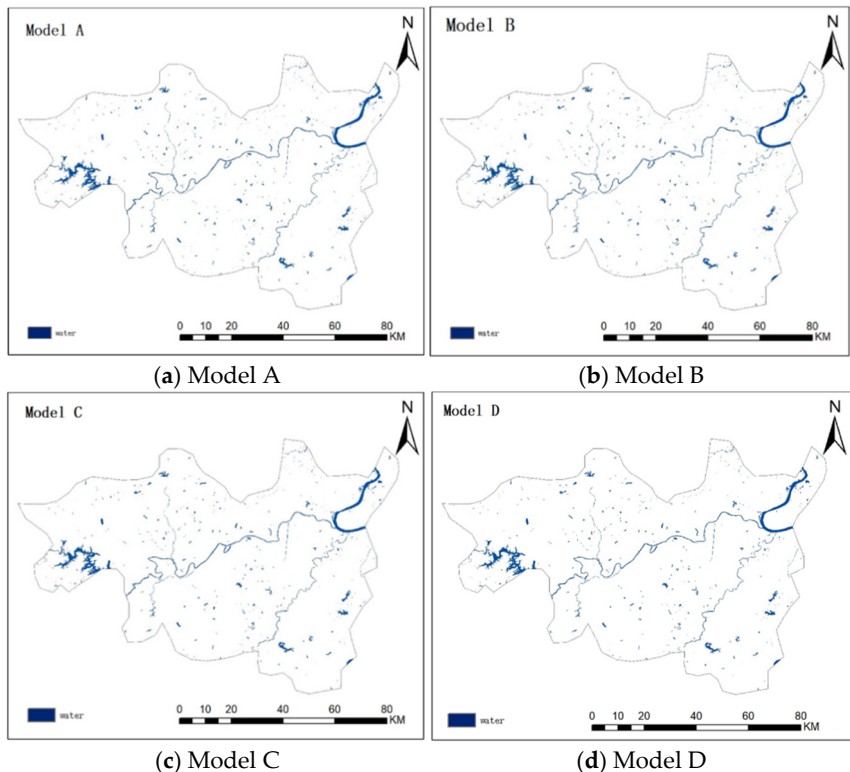

**Figure 3.** Accuracy comparison of four models (model A, model B, model C, and model D).

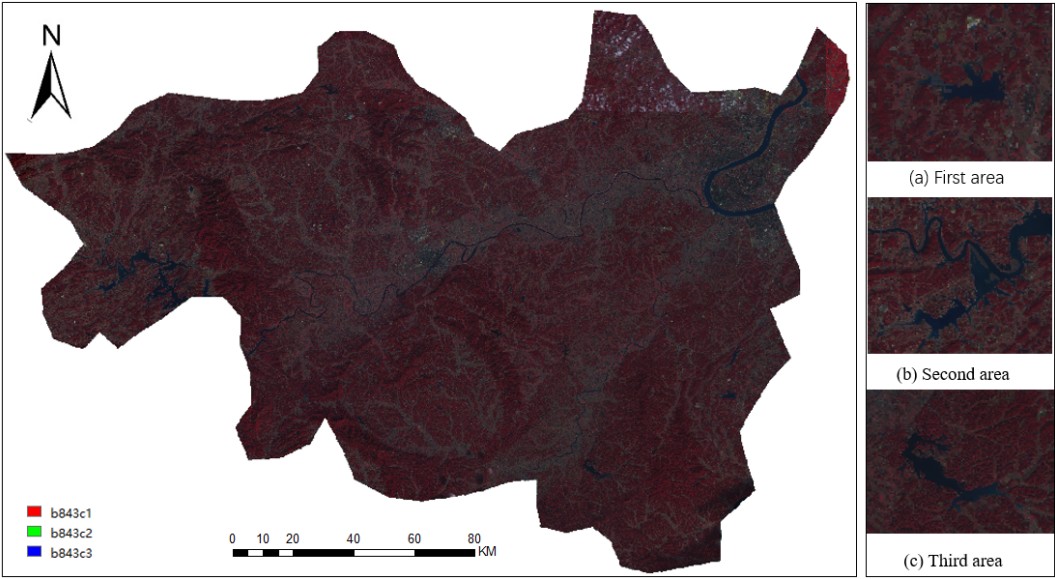

**Figure 4.** Standard false color image of Sentinel-2 data in Xiangtan City and three local area images in (**a**–**c**).

**Table 4.** Classification accuracy statistics of models A, B, C, and D.

| Types | Model A | | Model B | | Model C | | Model D | |
|---|---|---|---|---|---|---|---|---|
| | Producer's Accuracy (PA) | User's Accuracy (UA) | Producer's Accuracy (PA) | User's Accuracy (UA) | Producer's Accuracy (PA) | User's Accuracy (UA) | Producer's Accuracy (PA) | User's Accuracy (UA) |
| Water | 80.63 | 83.22 | 82.16 | 86.22 | 91.39 | 94.12 | 92.16 | 95.22 |
| Cultivation land | 85.43 | 79.26 | 88.52 | 81.53 | 89.21 | 82.46 | 91.25 | 84.53 |
| Forest | 90.45 | 92.76 | 92.87 | 93.98 | 93.07 | 94.19 | 93.98 | 95.06 |
| Grassland | 82.36 | 76.52 | 84.51 | 79.84 | 87.12 | 81.65 | 90.12 | 83.54 |
| Wetland | 80.15 | 73.28 | 85.62 | 80.17 | 87.45 | 81.56 | 90.22 | 93.14 |
| Building land | 82.03 | 73.49 | 80.41 | 71.29 | 83.46 | 79.02 | 86.52 | 81.39 |
| Others | 85.87 | 81.37 | 88.41 | 81.61 | 91.67 | 84.62 | 94.67 | 88.62 |
| The overall accuracy (OA)% | 81.24 | | 86.39 | | 89.71 | | 92.37 | |
| Kappa coefficient | 0.8016 | | 0.8092 | | 0.8114 | | 0.8116 | |

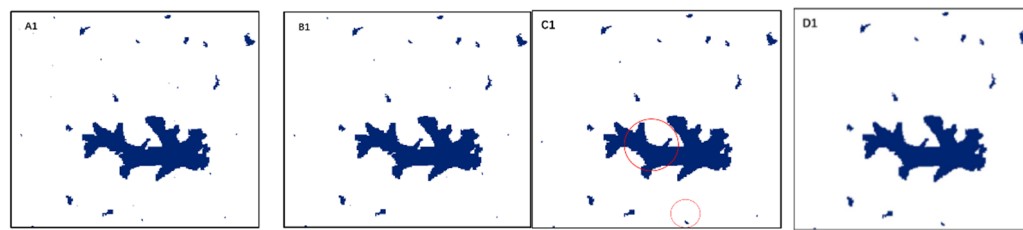

**Figure 5.** The classification result map of the four models corresponding to the first area. There is a lot of noise in (**A1**), the noise of (**B1**) is significantly reduced, the water body extracted by (**C1**) is more complete, and the water body extracted by (**D1**) has the best effect.

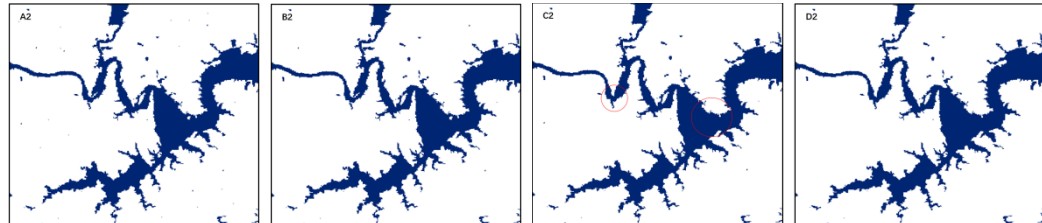

**Figure 6.** The classification result map of the four models corresponding to the second area. There is a lot of noise in (**A2**), the noise of (**B2**) is significantly reduced, the water body extracted by (**C2**) is more complete, and the water body extracted by (**D2**) has the best effect. Because adding the water body index factor to model C can improve the water body extraction accuracy.

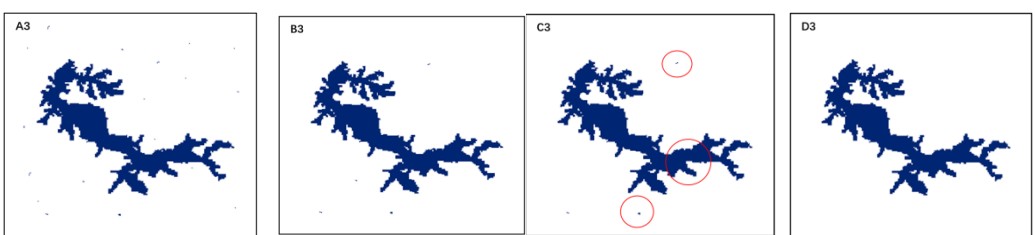

**Figure 7.** The classification result map of the four models corresponding to the third area. There is a lot of noise in (**A3**), the noise of (**B3**) is significantly reduced, the water body extracted by (**C3**) is more complete, and the water body extracted by (**D3**) has the best effect. Because adding the water body index factor to model C can improve the water body extraction accuracy. Adding terrain factors to Model D can reduce the misclassification of hillshade and water bodies.

Compared with model B, model C is visually clearer. The black and white dots are fewer and characteristics are more evenly spaced. (See Figures 5C1, 6C2 and 7C3). Meanwhile, the water index factors of NDWI, MNDWI, and LSWI are added. Among them, the improved water index factor MNDWI can strengthen open waters and dampen or even remove the black and white dots caused by clusters of buildings, vegetation, and soil. In addition, the clattered objects on the ground have been effectively merged. The extraction of linear objects on the ground is also more effective. The water extraction accuracy has increased from 82.16% to 91.39%. The overall accuracy has increased from 86.39 to 89.71%.

Compared with model C, because of the addition of terrain factors, model D's accuracy has been enhanced. The shadows of mountain areas and the wrong classification of waters are noticeably decreased (see Figure 7C3). The addition of topographic factors such as elevation, slope, and aspect can effectively reduce water misclassification.

### 4.2. The Evaluation of Characteristic Importance

One of the advantages of random forest classification method is its contribution and importance of evaluating characteristic variables [41]. Figure 6 shows the importance ranking of characteristic variables according to OOB error. It results in the best model being model D. According to the figure, the importance and contribution of each variable can be evaluated (See Figure 8). The original band data can mainly be seen in the front position of the ranking, so the normalized difference water index (MNDWI) and band B2 (Blue), normalized water index (NDWI), B4 (Red), B3 (Green), and band B5 (Vegetation Red-Edge 1) are improved. The importance of blue band (B2) is far higher than that of other bands. Because the blue band is sensitive to soil background reflection, it can effectively differentiate soil from vegetation. MNDWI has the noticeable effect of extracting water information [42]. For the classification of land use and land coverage, the characteristics of original band and the importance of MNDWI have been identified by many scholars. However, due to the different study areas and study time, the categories of objects and the growth state of plants are different. Therefore, the importance ranking differs. In general, they are among the former position [43]. Among the importance ranking, the following are traditional remote sensing dates (NDVI, B12, B11, NDI45, PSSRa, LSWI, and MSAVI). Most red-edge remote sensing data are among the behind position of the ranking. Additionally, the importance of terrain factors (elevation, slope, and aspect) is also lower.

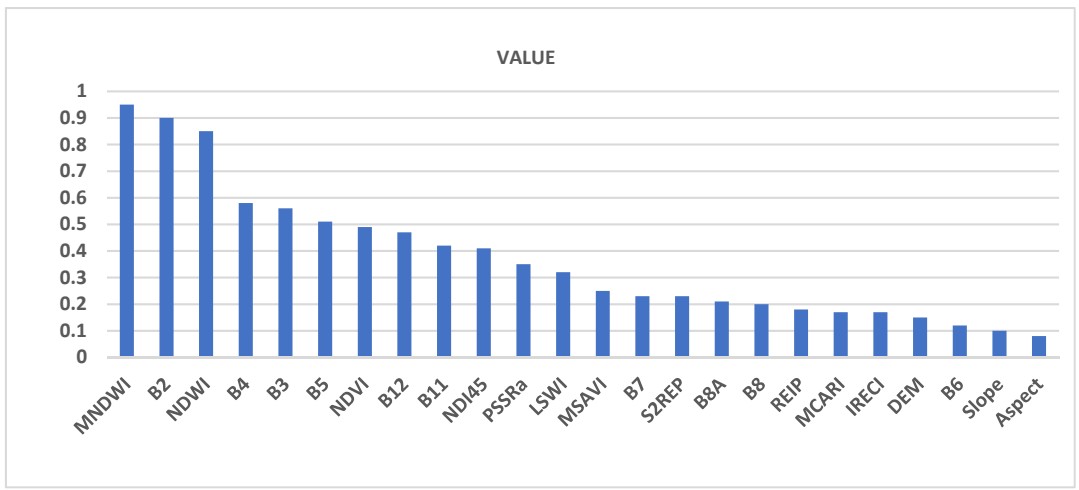

**Figure 8.** Characteristic variables importance value of model D.

The accuracy of the random forest classification results of 24 characteristic combinations is evaluated, and the relationship diagram between the number of characteristics and the classification accuracy was generated (see Figure 9). Conclusions can be reached that as characteristics increase, the variables with higher importance and contributions

are first used as the input data of random forest models. The correlation of characteristic variables is low. The overall accuracy and Kappa coefficient demonstrate a sharp increase. In the first period, as the visible band and red-edge index band are input into the random forest classification model, the overall accuracy and Kappa coefficient quickly increased, respectively, from 54.52% and 0.36 to 89.98% and 0.779. The tendency of growing classification accuracy is rapid. In the middle period, the increase fluctuates. When the number of characteristics reaches 16, the classification and Kappa coefficient reach the highest level, 92.57% and 0.8116, respectively. When characteristic numbers increase from 16 to 24, the data become redundant, and more time is needed due to the interconnectivity of different characteristics. The performance of classifier is worse than before. Therefore, the phenomenon of over-fitting of the classification results is in the trend of decreasing fluctuation of the classification accuracy. For this reason, in order to ensure the classification accuracy and improve the classification efficiency, the research carried out further feature variable selection and reduced the dimension of 24 feature variables. Two methods are mainly used for comparative analysis.

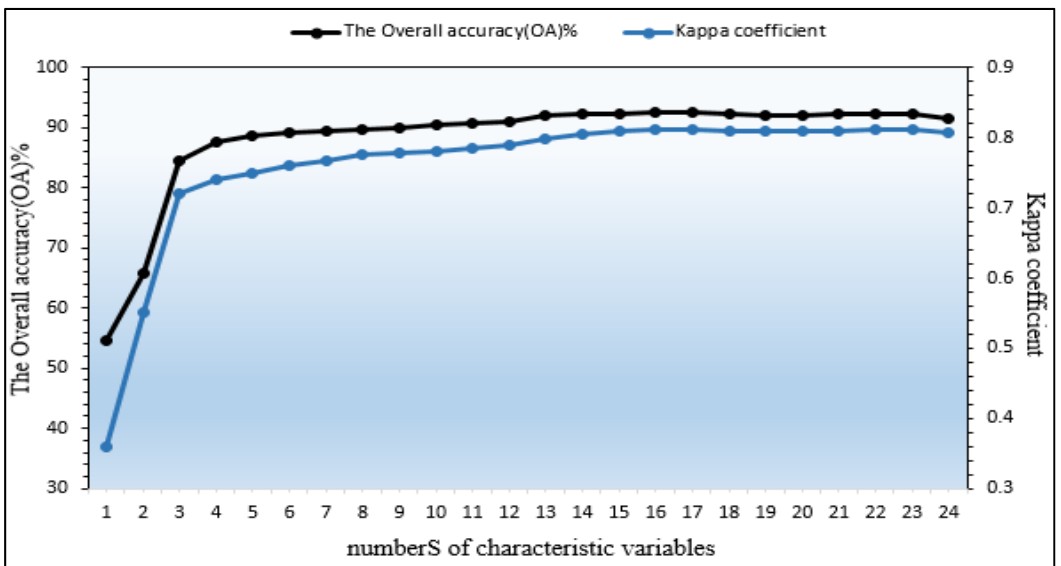

**Figure 9.** The relationship between the number of features and classification accuracy.

(1) Choose based on variable importance estimates. According to the importance of the feature variables of model D and the first 16 feature variables of model D (MNDWI, B2, NDWI, B3, B5, NDVI, B12, B11, NDI45, PSSRa, LSWI, B7, S2REP, B8A, B8, and REIP), the new model is recorded as RF_16.

(2) The 24 feature variables are screened by the Relief F algorithm. The Relief F algorithm is a feature selection algorithm that considers the interdependence of multiple variables and features. This algorithm can effectively deal with noisy multi-class and regression problems. The Relief F algorithm evaluates the classification ability of the feature through the "hypothesis interval", considering the class distance and the intra-class distance comprehensively. If the class distance is greater than the intra-class distance, the weight is increased; the weights are continuously updated through the class spacing and intra-class distance, and feature selection is performed according to the final calculated weights. The weight update formula is shown in Equation (3), assuming that the interval θ refers to when the sample category is kept unchanged. For the maximum distance that the classification decision surface can move, see Formula (4).

$$w_f^i = w_f^{i-1} + \frac{1}{n}\left\{ \frac{\sum_{c \neq class(x_i)} diff_f[x_i, M(x_i)]}{(n-1) \times m_{class(x_i)}} - \frac{diff_f[x_i, H(x_i)]}{m_{class(x_i)}} \right\} \qquad (6)$$

$$\theta = \frac{1}{2}[\|x - M(x)\| - \|x - H(x)\|] \tag{7}$$

In the formula, *diff* () is the distance between different samples, *n* is the number of samples, *f* is the feature of the evaluation, *i* is the randomly selected sample, and *M*(*x*), *H*(*x*) are the same and the closest to the sample *x*, respectively, and are adjacent sample points.

The Relief F algorithm is used to select the 24 feature variables of model D, and the first 16 variables that are more related to the target class are reserved, and the model with the same number as RF_16 is constructed, which is recorded as Relief-16. The variable combination model Relief-16 based on the Relief F algorithm dimensionality reduction, and the variable combination model RF_16 based on variable importance estimation dimensionality reduction are classified by random forest algorithm, and the algorithm execution time, classification accuracy, and Kappa coefficient are compared, see Table 5. The RF_16 algorithm is compared with Relief-16 and model D, and the overall classification accuracy is increased by 0.12% and 0.79%, respectively, and the Kappa coefficient is 0.0048 and 0.0108 higher than that of Relief-16 and model D, respectively. The lowest classification accuracy of cultivated land is 90.12%. In terms of the cultivated land classification accuracy of the Relief-16 model, the highest is 92.21%, which shows that the RF_16 algorithm is not the best cultivated land extraction model. However, the ground type with the highest classification accuracy of the RF_16 model is the water body, which is 94.16%. Compared with the Relief-16 algorithm and the model D water body extraction accuracy, the extraction accuracy is increased by 1.77% and 2%, respectively. The extraction accuracy of other ground object types is higher than 91%. The classification effect is generally better. Therefore, in comparison of the three models, the RF_16 model has the best effect in classifying land use information, especially in terms of water body extraction.

**Table 5.** Classification accuracy statistics of models RF_16, Relief-16, and model D.

| Types | RF_16 | | Relief-16 | | Model D | |
|---|---|---|---|---|---|---|
| | Producer's Accuracy (PA) | User's Accuracy (UA) | Producer's Accuracy (PA) | User's Accuracy (UA) | Producer's Accuracy (PA) | User's Accuracy (UA) |
| Water | 94.16 | 96.22 | 92.39 | 95.42 | 92.16 | 95.22 |
| Cultivation land | 90.12 | 88.53 | 92.21 | 85.46 | 91.25 | 84.53 |
| Forest | 93.87 | 94.98 | 93.17 | 94.19 | 93.98 | 95.06 |
| Grassland | 91.51 | 88.34 | 91.12 | 84.65 | 90.12 | 83.54 |
| Wetland | 93.62 | 90.17 | 92.45 | 91.56 | 90.22 | 93.14 |
| Building land | 91.41 | 89.29 | 89.56 | 85.02 | 86.52 | 81.39 |
| Others | 94.81 | 92.61 | 91.67 | 89.62 | 94.67 | 88.62 |
| The overall accuracy (OA)% | 93.16 | | 93.04 | | 92.37 | |
| Kappa coefficient | 0.8224 | | 0.8176 | | 0.8116 | |

The classification results and differences are shown in Figure 10. Figure 10a is a partial map of the classification effect of the Relief-16 model, and Figure 10b is a partial map of the same position of the classification effect of the RF_16 model. According to the comparison of areas 1, 3, 5, 6, and 7 in the figure, it can be seen that the extraction of ground objects from the classification results of the RF_16 model is more complete, reducing the misclassification of wetlands and woodlands. In areas 2 and 4, the misclassification of water bodies is reduced, and the water body extraction is more complete and continuous. At the same time, in area 4, it can be seen that based on the RF_16 model, the misclassification of building land and water bodies is reduced, and the overall classification effect is better, which further proves that the RF_16 model has strong adaptability in water body extraction.

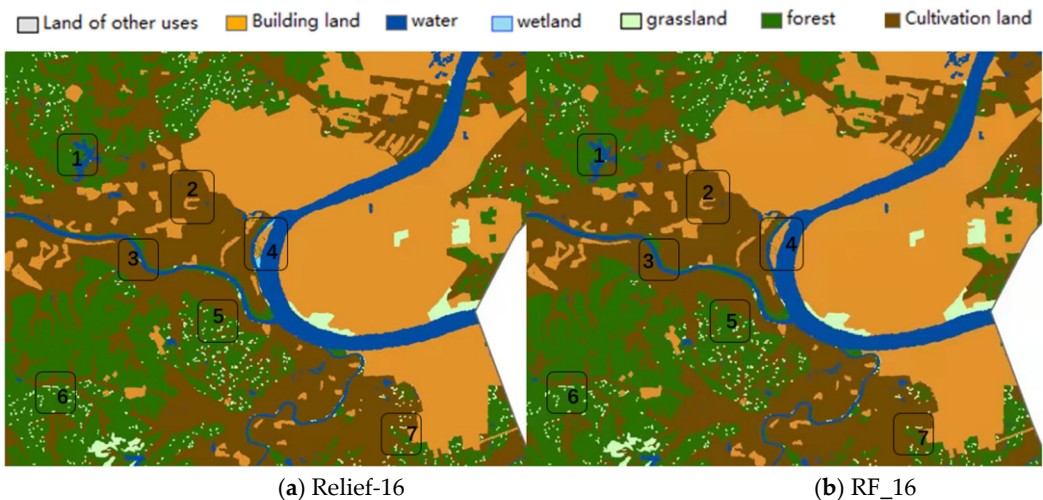

(**a**) Relief-16        (**b**) RF_16

**Figure 10.** Comparison of classification results of models RF_16 and Relief-16.

Figure 11 is the land use classification map of Xiangtan City obtained by a random forest algorithm based on the RF_16 model. It can be seen from this figure that the land use classification results of Xiangtan City based on the RF_16 model algorithm have good spatial feasibility, especially the extracted water body information, which is more complete and more accurate. It shows that adding red-edge band information in land use cover can improve the classification accuracy, while adding a vegetation index can effectively extract vegetation information, and adding a water body index factor can greatly improve the accuracy of water body extraction. The extracted water body information data is superimposed on the true color remote sensing image data of Xiangtan City, as shown in Figure 12, which can provide better information for future water body research, databases, and technical background support.

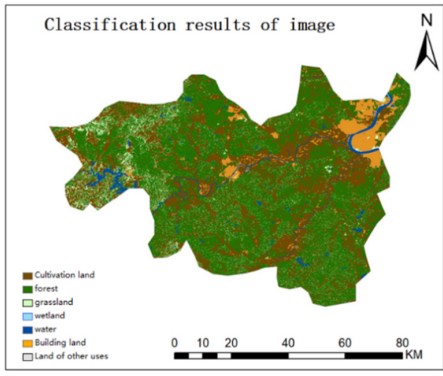

**Figure 11.** Classification results of image.

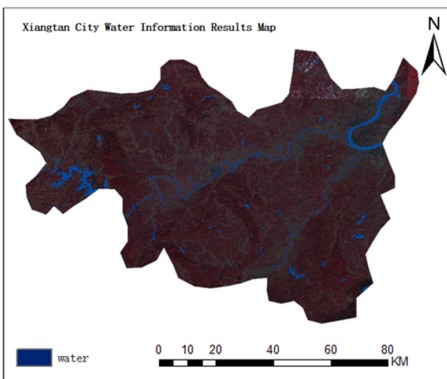

**Figure 12.** Xiangtan City Water Information Results Map.

### 4.3. Accuracy Evaluation

To further evaluate RF_16, the water extraction effect of the 16 model will be based on RF_16. Compare the water extraction results of the model with the water extraction results of NDWI, MNDWI, and NDVI, and evaluate the accuracy of the calculation results of these methods, respectively. See Table 6 for the accuracy evaluation of each method.

**Table 6.** Evaluation of extraction accuracy of water extraction by different methods.

| Methods | Kappa Coefficient | Missed Extraction Rate/% | False Extraction Rate /% | The Overall Accuracy (OA)% |
|---------|-------------------|--------------------------|--------------------------|----------------------------|
| NDVI    | 0.8046            | 2.86                     | 13.72                    | 89.22                      |
| NDWI    | 0.8112            | 1.94                     | 11.24                    | 90.12                      |
| MNDWI   | 0.8206            | 1.89                     | 10.56                    | 92.57                      |
| RF_16   | 0.8224            | 0.48                     | 7.22                     | 93.16                      |

Comprehensive analysis based on the RF_16 model method has the lowest missed detection rate and a low false extraction rate; the missed detection rate of NDVI is the highest, and the false extraction rate is also high; the missed detection rates of NDWI and MNDWI are the same, but the false extraction rate of mNDVI is lower than that of NDVI, indicating that the water extraction effect is better than that of NDVI, but based on the RF_16 model method, it has the lowest false extraction rate. Simultaneously, based on RF_16, the water extraction effect of the 16 model method is the best, but it is based on RF_. When extracting water bodies by the model method, the selection of samples takes time and energy, and the quality of samples will directly affect the quality of the RF-based water body. The extraction accuracy of the model, and so the selection of samples in water extraction, is very key, but the operation of the water index method is simple and saves time.

In order to reflect the effect of water body extraction by the RF_16 model concretely and clearly and compare it with the water body effect extracted by NDVI, NDWI, and MNDWI, we selected the water body extraction effect maps of three local areas in Xiangtan City. Table 7 shows the different water bodies in Xiangtan City and a schematic diagram of the results of the extraction method. It can be seen from the table that the water bodies extracted by NDVI, NDWI, and MNDWI obviously have certain missed detections and false detections. In particular, the NDVI normalized vegetation index extraction results of water bodies have the highest missed detection rate and the highest false detection rate. Especially, NDVI, NDWI, and MNDWI all mistakenly classify hill shadows as water bodies, while the RF_16 model does not. Comparative analysis of the four methods of water body extraction shows that the RF_16 model extracts the water body extraction results with the least noise, and the lowest missed detection rate and false detection rate. Therefore, comparing the four methods, the overall extraction accuracy of the RF_16 model is the highest.

**Table 7.** Comparison of different water extraction methods.

| Methods | Extraction Effect | First Area | Second Area | Third Area |
|---------|-------------------|------------|-------------|------------|
| NDVI |  |  |  |  |
| NDWI |  |  |  |  |
| MNDWI |  |  |  |  |
| RF_16 |  |  |  |  |

## 5. Discussion

With the continuous progress of remote sensing technology, the ability of remote sensing data acquisition has become stronger and stronger. The Sentinel-2A satellite remote sensing data has rich spectral bands and high spatial resolution, providing a multi-dimensional feature space for land cover remote sensing classification. However, the multi-dimensional feature space easily causes information redundancy during classification, which leads to the reduction of classification accuracy and algorithm running speed. Commonly used methods for dimensionality reduction or feature optimization include the ReliefF algorithm [44], principal component analysis [45], etc. Different methods are suitable for different datasets and scenarios, and a unified conclusion has not yet been formed. As an ensemble learning method, random forest has the characteristics of high efficiency and high accuracy. It can not only ensure high accuracy but also fast speed in medium- and high-resolution image classification and has the ability of feature selection.

While making full use of the rich spectral information and spatial information of Sentinel-2A data, and by reducing the dimensionality of the high-dimensional feature variable model and using the out-of-bag (OBB) error, the important feature combinations are prioritized and finally selected. The optimal combined model RF_16 includes 16 characteristic variables of MNDWI, B2, NDWI, B3, B5, NDVI, B12, B11, NDI45, PSSRa, LSWI, B7, S2REP, B8A, B8, and REIP. Among these spectral characteristic variables, the visible light band plays an important role in land use classification, and the vegetation red-edge band of Sentinel-2A also shows a higher value, which is consistent with the results obtained by Immitzer et al. [46]. Especially in the extraction of water bodies, adding a water body index factor and vegetation index factor can effectively improve the accuracy of water body extraction. Therefore, the importance ranking of MNDWI and NDWI factors is relatively high, and their importance is higher in wetland-type extraction.

Although the application research of Sentinel-2 data based on random forest in water body extraction in this paper has achieved considerable results, due to the limitations of various objective conditions, there are still many problems that need to be further explored, for example, compared with other classification methods, trying to use Sentinel-2 data in different phases further determines the accuracy of water body extraction based on feature-optimized random forest method. At the same time, although the classification method based on machine learning can achieve better classification accuracy, it still cannot solve the problem of mixed pixels. In future research, the decomposition of mixed pixels can be used for land use classification to extract water bodies. At the same time, comprehensive analysis using RF_16, the water extraction effect of 16 algorithm, is the best, but the limitation is that the selection of samples is time consuming and energy consuming. The operation of the water index method is simple and relatively time saving. The extraction method of water body can be selected according to the actual needs.

## 6. Conclusions

In order to improve the water body extraction accuracy of remote sensing images, this study is based on Sentinel-2 image data, and four kinds of variables are extracted, including traditional remote sensing data, red-edge remote sensing data, water body index factor, and terrain factor; a total of 24 characteristic variable indicators build four combined models with different input variables. The importance of different characteristic variables in random forest and the reliability and practical value of the importance of characteristic variables calculated by random forest are studied, and the importance of each variable is analyzed. The water body is extracted based on the RF_16 optimal combination model algorithm after feature optimization and the traditional Relief F feature screening algorithm. After analysis, it is determined that RF_16 is the optimal combination model for extracting water body information in the study area. The random forest classification results are involved in the subsequent accuracy evaluation. To extract the water body information in the study area, the results show that:

(1) When red-edge remote sensing data, water index factors, and terrain factors are added to traditional sensing data models in sequence, the overall accuracies are, respectively, 81.24%, 86.39%, 89.71%, and 92.37%; Kappa coefficients are, respectively, 0.8016, 0.8092, 0.8114, and 0.8116.

(2) The characteristic variables that have important influence on model accuracy, such as normalized difference water index (MNDWI), band B2 (Blue), normalized water index (NDWI), B4 (Red), B3 (Green), and band B5 (Vegetation Red-Edge 1), have been improved.

(3) The optimal combination model RF_16 has a water extraction accuracy of 93.16% and a Kappa coefficient of 0.8214. The overall accuracy is 0.12% better than the traditional Relief F algorithm. The RF_16 method based on the optimal combination model of random forest is an effective means to obtain high-precision water body information in the study area. It can effectively reduce the "salt and pepper effect" and the influence of mixed pixels such as water and shadows on the water extraction accuracy.

In conclusion, different feature variables of random forest have different contributions to the model and have different effects on the classification accuracy. The importance of the calculated feature variables is reliable, and the classification accuracy after screening important feature variables is improved, which gives it a good practical value. Feature-based optimization of the RF_16 model can play an important role in optimizing the extraction of water body information, and also provide a scientific basis and decision support for the management of water body information in the future.

**Author Contributions:** Y.W. conceived and designed the study. Z.J. wrote the first draft, performed the data analysis, and collected all the study data. G.Z. and X.W. provided critical insights in editing the manuscript. All authors have read and agreed to the published version of the manuscript.

**Funding:** This work was supported in part by the Science and Technology Innovation Platform and Talent Plan Project of Hunan Province under Grant 2017TP1022 and in part by the Hunan Province Emergency Management Technology Project (2020YJ007).

**Institutional Review Board Statement:** Not applicable.

**Informed Consent Statement:** Not applicable.

**Data Availability Statement:** Not applicable.

**Acknowledgments:** The author would like to thank the teachers for their careful guidance and the students for their kind help.

**Conflicts of Interest:** The authors declare no conflict of interest.

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
