# Peer review of "Water Information Extraction Based on Multi-Model RF Algorithm and Sentinel-2 Image Data"

_sustainability, doi:10.3390/su14073797_

Round 1

Reviewer 1 Report

This research conceptually, correct. However, this is based on specific case study and very limited dataset. This is not giving reliable contribution to water sustainability and assessment.

Author Response

Response to the Review Comments

We gratefully than the editor and all reviewers for their time spend making their constructive remarks and useful suggestions, which has significantly raised the quality of the manuscript and has enable us to improve the manuscript. Each suggested revision and comment, brought forward by the reviewers was accurately incorporated and considered. Below the comments of the reviewers are response point by point and the revisions are indicated.

Reviewer1

General Comments: This research conceptually, correct. However, this is based on specific case study and very limited dataset. This is not giving reliable contribution to water sustainability and assessment.

Reply:We gratefully appreciate for your valuable comment. The main research purpose of this work is to solve the problem that traditional water extraction methods cannot meet the needs of practical applications due to the rich spatial information of Sentinel-2 multispectral satellite imagery remote sensing data. In this paper, a random forest-based RF_16 optimal combination model algorithm is proposed to extract water bodies. The research process uses Sentinel-2 multispectral satellite images and DEM data as the basic data, and collected 24 characteristic variable indicators (B2, B3, B4, B8, B11, B12, NDVI, MSAVI, B5, B6, B7, B8A, NDI45 , MCARI, REIP, S2REP, IRECI, PSSRa, NDWI, MNDWI, LSWI, DEM, SLOPE, SLOPE ASPECT), and constructed 4 combination models with different input variables. After analysis, it was determined that RF_16 was the optimal combination for extracting water body information in the study area. Model. The results show that: (1) The characteristic variables that have an important impact on the accuracy of the model are the improved normalized difference water index (MNDWI) and, band B2 (Blue), normalized water index (NDWI), B4 (Red), B3 (Green), band B5 (Vegetation Red Edge 1); (2) The water extraction accuracy of the optimal combined model RF_16 can reach 93.16%, and the kappa coefficient is 0.8214. The overall accuracy is 0.12% better than the traditional Relief F algorithm. The RF_16 method based on the optimal combination model of random forest is an effective means to obtain high-precision water body information in the study area. It can effectively reduce the "salt and pepper effect" and the influence of mixed pixels such as water and shadows on the water extraction accuracy.

Feature-based selection of the RF_16 model can play an important role in optimizing the extraction of water body information, which has a very good practical value, and also provides scientific basis and decision support for the management of water body information in the future.

Each comment will be directly addressed regarding the modified manuscript with changes highlighted in yellow.

Thank you again for your positive and constructive comments and suggestions on our manuscript.

We hope you will find our revised manuscript acceptable for publication.

Reviewer 2 Report

Dear Authors,

I carefully evaluated your manuscript “Water Information Extraction Based on Multi-model RF algorithm and Sentinel-2 Image Data.” The topic is within my area of expertise and I consider the topic interesting and well within the scopes of the journal. However, I identified some flaws in your manuscript for which I strongly recommend major revision before being considered again for publication. You can find my comments below.

  1. L21: I think the authors directly copied their thesis abstract. Instead of mentioning “study”, the author uses thesis. I am little offended by this carelessness. Please correct the word: study instead of thesis. Follow the same for other section.
  2. The abstract should include the main research gap that this study intends to address.
  3. Concluding remarks are not available in the abstract. Please stress this in the manuscript.
  4. What is the novelty of your work? Models, parameters or analysis? Please stress it.
  5. Why do the authors choose the mentioned models? This is very important matter to discuss in the manuscript.
  6. The background of the study lacks literature related to uncertainties in modeling. The authors only compared the results of various performance assessment indices. I do not see any attempt to quantify the spatial agreement among resulting maps produced using different models.
  7. Are there any collinearity effect among characteristic variables? If yes, then the authors should assess the multicollinearity effect and report to the results.
  8. What is justification of selecting 24 characteristic indicators? Stress it in the manuscript.
  9. The authors require to analyze their results based on other existing literature. I merely see these in the discussion part.
  10. The conclusion section needs uncertainty assessment, limitation of the existing approach, and policy implications of the study.
  11. Substantial editing is needed not only to correct the numerous grammar mistakes which exist throughout the manuscript but also to improve readability. Several sentences throughout the manuscript are very difficult to understand/interpret.

Author Response

Response to the Review Comments

We gratefully than the editor and all reviewers for their time spend making their constructive remarks and useful suggestions, which has significantly raised the quality of the manuscript and has enable us to improve the manuscript. Each suggested revision and comment, brought forward by the reviewers was accurately incorporated and considered. Below the comments of the reviewers are response point by point and the revisions are indicated.

Reviewer2

1.Comment: L21: I think the authors directly copied their thesis abstract. Instead of mentioning “study”, the author uses thesis. I am little offended by this carelessness. Please correct the word: study instead of thesis. Follow the same for other section.

1.Reply:Sorry to offend you by my carelessness, I have changed all "thesis" to "research" in the article. And modified the summary section in my research.

2.Comment: The abstract should include the main research gap that this study intends to address.

  1. Reply:Thank you for your valuable comments. The main research gap solved in this study is the problem that traditional water extraction methods cannot meet the needs of practical applications due to the rich spatial information of Sentinel-2 multispectral satellite imagery remote sensing data. This study proposes a random forest-based RF_16 optimal combination model algorithm to extract water bodies. The optimal combination model RF_16 has a water extraction accuracy of 93.16% and a kappa coefficient of 0.8214. The overall accuracy is 0.12% better than the traditional Relief F algorithm. At the same time, the results of RF_16 optimal combination model water extraction can effectively reduce the "salt and pepper effect" and the influence of mixed pixels such as water and shadows on the water extraction accuracy.

3.Comment: Concluding remarks are not available in the abstract. Please stress this in the manuscript.

3.Reply: Thank you for your valuable comments, and I have added the concluding remarks to my summary.

4.Comment: What is the novelty of your work? Models, parameters or analysis? Please stress it.

4.Reply: Generally speaking, water body information extraction research is usually a single method, such as object-oriented water body extraction, or water body information extraction using water body index. The novelty of this work In this study, a random forest-based RF_16 optimal combination model algorithm is proposed to extract water bodies. Results Compared with the traditional Relief F, it is better to effectively suppress or even eliminate the noise spatial agglomeration effect of buildings, vegetation and soil, so that the finely classified objects can be effectively merged, and the extraction effect of linear objects is more obvious. The water extraction effect is also better.

5.Comment: Why do the authors choose the mentioned models? This is very important matter to discuss in the manuscript.

5.Reply: Although model D has high accuracy, it is prone to overfitting due to too many feature variables. Therefore, the redundant feature variables are removed by the dimensionality reduction method, and the random forest classification and comparison are performed again after feature selection by variable importance estimation and Relief F method, and it is found that the model extraction accuracy of the optimal combination model RF_16 is higher. Therefore, RF_16 is determined as the optimal combination model for extracting water body information in the study area.

6.Comment: The background of the study lacks literature related to uncertainties in modeling. The authors only compared the results of various performance assessment indices. I do not see any attempt to quantify the spatial agreement among resulting maps produced using different models.

6.Reply: In order to compare the specific differences in the results obtained by different models, we mainly selected three local area maps of the study area for comparison, as shown in Figure 4-Figure 7. The three local areas correspond to the classification effect maps under the four models. And I discuss those differences in the results in detail in the conclusion section. At the same time, the comparison chart of the optimal combination model RF_16 and the Relief F algorithm to extract the water body information in the study area is compared.

7.Comment: Are there any collinearity effect among characteristic variables? If yes, then the authors should assess the multicollinearity effect and report to the results.

7.Reply: Thank you for your valuable questions. I used SPSS software to perform a stepwise regression analysis (Stepwise Regression) on the 24 characteristic variables, and the collinearity diagnosis of the 24 characteristic variables. The Tolerance of the 24 characteristic variables displayed in the "regression coefficient table" (Tolerance) are all greater than 0.2 and VIF (variance expansion factor) are all less than 10. At the same time, the eigenvalues displayed in the "collinearity diagnosis" result table are not close to 0, and are far greater than 0, and the eigenvalues are all less than 10 . This is consistent with the results in the regression coefficient table. It shows that there is no collinear relationship between the variables. We have put these two tables in the appendix for your review. Hope that resolves your concerns.

8.Comment: What is justification of selecting 24 characteristic indicators? Stress it in the manuscript.

8.Reply: Thanks for your valuable comments, I have included the rationale for choosing the 24 feature metrics in the manuscript. I think I still need to explain here. Compared with traditional remote sensing data, Sentind-2A remote sensing data has added 4 red-edge bands (B5, B6, B7, B8A) that are closely related to chlorophyll content. The use of coverage classification, wetland extraction, crop identification research, and the use of red edge bands (including NDI45, MCARI, REIP, S2REP, IRECI, PSSRa, etc.) have achieved good results. At the same time, a large number of studies have shown that the introduction of terrain factors can significantly improve the classification accuracy of land use or water body extraction, and the introduction of water body index factors can further improve the accuracy of water body extraction.

9.Comment: The authors require to analyze their results based on other existing literature. I merely see these in the discussion part.

9.Reply: Thank you for your valuable comments, and I supplement the relevant analysis results in the first part of the research background.

  1. Comment: The conclusion section needs uncertainty assessment, limitation of the existing approach, and policy implications of the study.

10.Reply: Thank you for your valuable comments, I have supplemented the Discussion section.

11.Comment: Substantial editing is needed not only to correct the numerous grammar mistakes which exist throughout the manuscript but also to improve readability. Several sentences throughout the manuscript are very difficult to understand/interpret.

11.Reply: I'm very sorry for the trouble you have caused you to read. I read it carefully and revised the grammar-related issues.

Each comment will be directly addressed regarding the modified manuscript with changes highlighted in yellow.

Thank you again for your positive and constructive comments and suggestions on our manuscript.

We hope you will find our revised manuscript acceptable for publication.

Reviewer 3 Report

This manuscript aims to analyze the uses senti nel-2 satellite multi-spectral images and DEM data to extract the water information of Hunan province based on random forest algorithm. Based on my review, the manuscript has  comments that need to consider before resubmission process. The following comments are suggested to improve the quality of this work.

General comments:

  • In this study, authors used 10000 sample points by ArcGIS random sampling tool. The samples are divided into two parts, 70% of them are used as the training sample set for modeling, and the remaining 30% used for validation. How did the authors deal with the overfitting values?
  • The authors applied the random forest in this work but the main model parameters that used should be indicated. What is the fitness function that applied to this algorithm? Also, the flowchart of the presented model is necessary to be indicated.
  • For this work, different parameters have been inserted as inputs to the RF algorithm, but authors need to present the sensitivity of the used input parameters and observe why they specifically used these parameters?
  • The results are relevant, but the reasons of these actions for the influence of different parameters are still vague, i.e., Figs. 9-11 you mentioned the relationship between the number of features and classification accuracy and results of image, but the explanations are still superficial!
  • Authors should observe how the analyses of the remote sensing figure, based on the Sentinel-2 image data can be benefit to the practical engineering projects.

Specific comments

Based on the presented abstract, the main aim from this study should be clearly presented. Otherwise, the abstract is long with several ineffective information, e.g., lines 14-18, etc. Abstract should be written using the following sequence: 1) the gap or the problem investigated, 2) methodology used to solve the problem, and 3) key points of conclusions.

Line 44-50: authors need to indicate for the recent studies that applied machine learning algorithms based on different applications, e.g.,

Shaban et al. 2021a,b. Journal of Cleaner Production, 327, 129355.
https://doi.org/10.1016/j.jclepro.2021.129355.
  Resources conservation recycling. 169, 2021, 105443.
https://doi.org/10.1016/j.resconrec.2021.105443.

Line 70-100: The different between this study and current literature should be clearly indicated.

The word meaning of “thesis” is suggested to be changed to become “paper” or “work”.

Line 144-145: Sentinel-2 data pre-processing mainly includes atmospheric correction, image registration and image fusion. How were these data collected?

Line 157-160: authors should indicate the source reference of these information.

Line 196: The Spectral reflectance curve of ground objects corresponding to each characteristic variable has been shown in Figure 2. How this figure can be benefit to the reader.

Line 219: in this study, author used 10000 sample points by ArcGIS random sampling tool. How were these samples collected?

Line 225: why the authors specifically used 70% of data for training and 30% for testing?

Author Response

Response to the Review Comments

We gratefully than the editor and all reviewers for their time spend making their constructive remarks and useful suggestions, which has significantly raised the quality of the manuscript and has enable us to improve the manuscript. Each suggested revision and comment, brought forward by the reviewers was accurately incorporated and considered. Below the comments of the reviewers are response point by point and the revisions are indicated.

Reviewer3

General comments:

1.Comment: In this study, authors used 10000 sample points by ArcGIS random sampling tool. The samples are divided into two parts, 70% of them are used as the training sample set for modeling, and the remaining 30% used for validation. How did the authors deal with the overfitting values?

 1.Reply:Thank you for this valuable question, about dealing with overfit values, maybe I didn't make it clear. In this work, I use the random forest algorithm itself, which has good anti-fitting performance. Due to the combination of trees, random forest can deal with nonlinear data. Due to randomness, it is very effective in reducing the variance of the model, so Random forest generally does not need additional pruning, that is, it can achieve better generalization ability and anti-overfitting ability (Low Variance).

The phenomenon of overfitting in this paper is mainly due to the fact that when the number of features increases from 16 to 24, data redundancy and time-consuming increase due to the correlation between features, which reduces the performance of the classifier, resulting in the phenomenon of classification overfitting. Therefore, in order to ensure the classification accuracy and improve the classification efficiency, the research carried out further feature variable selection, and reduced the dimension of 24 feature variables. Two methods are mainly used for comparative analysis.

1) Choose based on variable importance estimates. According to the importance of the feature variables of model D, the first 16 feature variables of model D (MNDWI, B2, NDWI, B3, B5, NDVI, B12, B11, NDI45, PSSRa, LSWI, B7, S2REP, B8A, B8, REIP), the new model is recorded as RF_16.

2) The 24 feature variables are screened by the Relief F algorithm. The Relief F algorithm is a feature selection algorithm that considers the interdependence of multiple variables and features. This algorithm can effectively deal with noisy multi-class and regression problems.

Meanwhile, I have added content to my research (L373-L392).

2.Comment: The authors applied the random forest in this work but the main model parameters that used should be indicated. What is the fitness function that applied to this algorithm? Also, the flowchart of the presented model is necessary to be indicated.

  1. Reply: Thank you for your valuable comments. In order to evaluate the classification error of the random forest algorithm, we mainly use the generated random forest classifier to classify the data. For the accuracy estimation, when each sample belongs to the OBB sample, its votes will be counted each time The majority vote will determine the classification category. Since the OBB sample does not participate in the establishment of the decision tree, it can be used to estimate the prediction error, use the OBB error to evaluate the model performance and quantify the importance of each feature variable. This is consistent with research by Aniruddha Ghosh and other researchers. In addition, I have put the main flow chart of the original work in the appendix section for your convenience.

3.Comment: For this work, different parameters have been inserted as inputs to the RF algorithm, but authors need to present the sensitivity of the used input parameters and observe why they specifically used these parameters?

3.Reply: Thank you for your valuable comments. In this work, in order to improve the classification accuracy, the Python grid search cross-validation tool (GridSearchCV) is used for parameter optimization, and the optimal parameter combination is determined. The optimization range of mtry parameters of model A is (2, 3, 4, 5), the optimization range of mtry parameters of model B is (3, 5, 7, 9), and the optimization range of mtry parameters of model C is (5, 10, 15), the mtry parameter optimization range of model D is (6, 12, 18); the ntree parameter optimization range is uniformly (25, 50, 75, 100). You can see these in my research.

4.Comment: The results are relevant, but the reasons of these actions for the influence of different parameters are still vague, i.e., Figs. 9-11 you mentioned the relationship between the number of features and classification accuracy and results of image, but the explanations are still superficial!

4.Reply: Thank you for your valuable comments, I have added the explanation about the image: the classification results and differences are shown in Figure 10, 10(a) is a partial map of the classification effect of the Relief-16 model, and 10(b) is the classification effect of the RF_16 model. Part of the same position picture. According to the comparison of areas 1, 3, 5, 6, and 7 in the figure, it can be seen that the extraction of ground objects from the classification results of the RF_16 model is more complete, reducing the misclassification of wetlands and woodlands. In areas 2 and 4, the misclassification of water bodies is reduced, and the water body extraction is more complete and continuous. At the same time, in area 4, it can be seen that based on the RF_16 model, the misclassification of building land and water bodies is reduced, and the overall classification effect is better, which further proves that the RF_16 model has strong adaptability in water body extraction. Figure 11 shows the land use classification map of Xiangtan City obtained by random forest algorithm based on the RF_16 model. The extracted water body information data is superimposed on the true color remote sensing image data of Xiangtan City, as shown in Figure 12, which can provide better information for water body research in the future. data base and technical background support.

5.Comment: Authors should observe how the analyses of the remote sensing figure, based on the Sentinel-2 image data can be benefit to the practical engineering projects.

5.Reply: Thank you for your valuable comments. I have added corresponding content in the results and discussion sections. For example, the RF_16 method based on the optimal combination model of random forest in this study is an effective means to obtain high-precision water body information in the study area. It can effectively reduce the "salt and pepper effect" and the influence of mixed pixels such as water and shadows on the water extraction accuracy.

Specific comments

  1. Comment:Based on the presented abstract, the main aim from this study should be clearly presented. Otherwise, the abstract is long with several ineffective information, e.g., lines 14-18, etc. Abstract should be written using the following sequence: 1) the gap or the problem investigated, 2) methodology used to solve the problem, and 3) key points of conclusions.

1.Reply: Thanks for your valuable comments, I have revised the summary section. You can see my edits in the summary.

2 Comment:Line 44-50: authors need to indicate for the recent studies that applied machine learning algorithms based on different applications, e.g.,

2.Reply: Thank you for your valuable comments, I have added the references you listed to my research, please check.

3.Comment:Line 70-100: The different between this study and current literature should be clearly indicated.

3.Reply: Thanks for your valuable comments, I have added the content to my research and you can view it in my research.

4 Comment:The word meaning of “thesis” is suggested to be changed to become “paper” or “work”

4.Reply: Thank you for your valuable input, it's really important, I've changed all " thesis " to "work".

5.Comment:Line 144-145: Sentinel-2 data pre-processing mainly includes atmospheric correction, image registration and image fusion. How were these data collected?

5.Reply: Thanks for your valuable comments, I am already adding data sources and other relevant information to my research. For your convenience, the added sections have also been marked in yellow in the study.

6.Comment:Line 157-160: authors should indicate the source reference of these information.

6.Reply: Thank you for your valuable comments, I have added the sources of this information in the research, and for your convenience, the added parts have also been marked in yellow in the research.

7.Comment:Line 196: The Spectral reflectance curve of ground objects corresponding to each characteristic variable has been shown in Figure 2. How this figure can be benefit to the reader.

7.Reply: Thank you for your valuable comments. The ground object Popper curve is mainly to reflect the prominent role of the red edge band in land use classification. So it was included in our study.

8.Comment:Line 219: in this study, author used 10000 sample points by ArcGIS random sampling tool. How were these samples collected?

8.Reply: Thank you for your valuable comments. These data are collected by: using ArcGIS10.4 software to generate a grid of 30 m × 30 m; (2) performing spatial overlay analysis on the grid and the current land use vector data, and the screening area is equal to 100 m2 (3) Convert vector data to point data, and extract about 300,000 points in total. On this basis, in order to improve the computing efficiency of random forest, 10,000 sample points were selected by ArcGIS random sampling tool, and the selected sample points fully covered the study area and met uniform distribution. In order to further ensure the accuracy of the data, the sample points were imported into Google Earth, outliers were eliminated, and finally 7226 sample points were retained, including 1390 water areas, 1849 cultivated land, 2987 forest land, 219 grassland, 156 wetland, and 249 other land. There are 376 construction sites. You can see these in the Sampling section of my work.

9.Comment:Line 225: why the authors specifically used 70% of data for training and 30% for testing?

9.Reply: Thank you for your valuable comments. In order to test the classification effect of each model, we divide the sample points into two parts in this work, and 70% of the sample data is used as the training sample set for modeling. The remaining 30% is used as a validation set, which does not participate in model construction, and is used to evaluate the accuracy of the classification effect of each model. this is very important.

 Each comment will be directly addressed regarding the modified manuscript with changes highlighted in yellow.

Thank you again for your positive and constructive comments and suggestions on our manuscript.

We hope you will find our revised manuscript acceptable for publication.

Round 2

Reviewer 2 Report

The authors have adequately addressed my comments. I recommend publication of the revised manuscript.

Author Response

Response to the Review Comments

We gratefully than the editor and all reviewers for their time spend making their constructive remarks and useful suggestions, which has significantly raised the quality of the manuscript and has enable us to improve the manuscript. Each suggested revision and comment, brought forward by the reviewers was accurately incorporated and considered.

Reviewer2

  1. Comments and Suggestions for Authors: The authors have adequately addressed my comments. I recommend publication of the revised manuscript.
  2. Reply: Thank you for acknowledging our research work, because of your suggestions, the revised article is better and readers can get more valuable information.

Thank you again for your positive and constructive comments and suggestions on our manuscript.

We hope you will find our revised manuscript acceptable for publication.

Reviewer 3 Report

My previous comments have been clearly considered and i think the manuscript can be accepted.

Author Response

Response to the Review Comments

We gratefully than the editor and all reviewers for their time spend making their constructive remarks and useful suggestions, which has significantly raised the quality of the manuscript and has enable us to improve the manuscript. Each suggested revision and comment, brought forward by the reviewers was accurately incorporated and considered.

Reviewer3

  1. Comments and Suggestions for Authors:My previous comments have been clearly considered and i think the manuscript can be accepted.
  2. Reply: Thank you for acknowledging our research work, because of your suggestions, the revised article is better and readers can get more valuable information.

Thank you again for your positive and constructive comments and suggestions on our manuscript.

We hope you will find our revised manuscript acceptable for publication.
